# The Determination of Mitochondrial Mass Is a Prerequisite for Accurate Assessment of Peripheral Blood Mononuclear Cells’ Oxidative Metabolism

**DOI:** 10.3390/ijms241914824

**Published:** 2023-10-02

**Authors:** Belay Tessema, Janine Haag, Ulrich Sack, Brigitte König

**Affiliations:** 1Institute of Clinical Immunology, Faculty of Medicine, University of Leipzig, 04103 Leipzig, Germany; Urich.Sack@medizin.uni-leipzig.de; 2Department of Medical Microbiology, College of Medicine and Health Sciences, University of Gondar, Gondar P.O. Box 196, Ethiopia; 3Magdeburg Molecular Diagnostics GmbH & Co. KG, 39104 Magdeburg, Germany; mmd@mmd-web.de (J.H.); Brigitte.Koenig@medizin.uni-leipzig.de (B.K.); 4Institute of Medical Microbiology and Virology, Faculty of Medicine, University of Leipzig, 04103 Leipzig, Germany

**Keywords:** mitochondrial mass, cellular stress assay, peripheral blood mononuclear cells, seasonal variation

## Abstract

Mitochondria are responsible for ATP synthesis through oxidative phosphorylation in cells. However, there are limited data on the influence of mitochondrial mass (MM) in the adequate assessment of cellular stress assay (CSA) results in human peripheral blood mononuclear cells (PBMCs). Therefore, the aim of this study was to determine MM in PBMCS and assess its influence on the results of CSA measurements. Blood samples were collected and sent to the laboratory for MM and CSA measurements during different seasons of the year. The mitochondrial mass was determined based on the mtDNA:nDNA ratio in PBMCs using quantitative real-time PCR (qRT-PCR). CSA was measured using Seahorse technology. The MM was significantly lower during summer and autumn compared to winter and spring (*p* < 0.0001). On the contrary, we found that the maximal respiration per mitochondrion (MP) was significantly higher in summer and autumn compared to winter and spring (*p* < 0.0001). The estimated effect of MM on mitochondrial performance was −0.002 pmol/min/mitochondrion (*p* < 0.0001) and a correlation coefficient (r) of −0.612. Similarly, MM was negatively correlated with maximal respiration (r = −0.12) and spare capacity (in % r = −0.05, in pmol/min r = −0.11). In conclusion, this study reveals that MM changes significantly with seasons and is negatively correlated with CSA parameters and MP. Our findings indicate that the mitochondrial mass is a key parameter for determination of mitochondrial fitness. Therefore, we recommend the determination of MM during the measurement of CSA parameters for the correct interpretation and assessment of mitochondrial function.

## 1. Introduction

Mitochondria are the intracellular organelles responsible for ATP synthesis through oxidative phosphorylation in cells. During normal physiological conditions, a small fraction of oxygen used by mitochondria is converted to reactive oxygen species (ROS) such as superoxide anions and H_2_O_2_ [1]. Mitochondrial respiratory function has also been demonstrated to decline in various human tissues during the aging process [2,3]. This is thought to be caused, e.g., by oxidative damage and the mutation of mitochondrial DNA (mtDNA) in the somatic tissues of aged individuals [3].

Mitochondrial dysfunction can promote changed energy expenditure and systemic inflammation that modifies susceptibility to energy-related pathologies associated with oxidative stress, such as obesity, cancer, and diabetes [4,5]. Genetic variations, either nuclear or mitochondrial, can also result in lower mitochondrial mass (MM) or function, which are exacerbated by aging, exposure to environmental toxins, lifestyle, and disease risk factors [6,7]. Importantly, normal genetic variation within mtDNA can be associated with changes in mitochondrial function and disease susceptibility that will be intertwined with cellular metabolism and inflammation [7,8,9,10].

The Seahorse technology is a milestone in enabling mitochondrial bioenergetics to make its way into routine medical diagnostics with high throughput [11,12,13]. This device allows easy and quick determination of oxygen consumption and extracellular acidification. Various assay parameters that can be determined by this technology help the user disentangle the metabolic situation in many cell types of the human body. One of the most important parameters obtained from the Seahorse assay is the spare capacity. The spare capacity is considered the increase in oxygen consumption after decoupling the respiratory chain, relative to the basal respiration [12,14]. The strength of this respiration, however, is dependent on various factors, ranging from the integrity of the mitochondrial membrane, the quality of the proteins of the respiratory chain, and the spatial orientation of these complexes [12,13,14,15,16,17,18].

Moreover, the presence of the necessary cofactors and the efficiency of various metabolic pathways leading to the mitochondrial respiratory chain are additional prerequisites for mitochondria function [19,20]. Besides these obvious parameters, even the mitochondrial mass might have an influence on the intensity of the spare capacity. In this regard, it was reported that the mitochondrial mass is related to the outbreak of symptoms in Leber’s hereditary optic neuropathy (LHON) [21]. The higher the number of mtDNA copies, the lower the detection of LHON-related symptoms, such as blindness.

In our previous study, we demonstrated that the results of the cellular stress assay (CSA) in peripheral blood mononuclear cells (PBMCs) with Seahorse technology are dependent on various parameters, such as the PBMC isolation method, season, and age [22]. This finding, among other previous reports, suggests the importance of further investigation for additional factors that might influence the interpretation of Seahorse CSA results for better diagnosis of mitochondrial dysfunctions. We believe that the analysis of the Seahorse-based CSA is incomplete as long as the number of mitochondria that actually contribute to the observed effects is unknown. We hypothesized that an increase in mitochondrial mass leads to an increase in maximal respiration per mitochondrion, also called mitochondrial performance (MP), and spare capacity. However, to the best of our knowledge, there are limited data on the influence of MM on the adequate assessment of CSA results in PBMCs. Therefore, the aim of this study was to determine the MM in PBMCs and to assess its influence on the results of CSA.

## 2. Results

### 2.1. The Effect of Season on the Mitochondrial Mass (mtDNA:nDNA Ratio)

The mtDNA:nDNA ratio was varied in different seasons of the year. The mtDNA:nDNA ratio was significantly lower in summer than in spring (*p* = 0.0003) and winter (*p* = 0.0056). Similarly, the mtDNA:nDNA ratio was significantly lower in autumn than in spring (*p* = 0.0009) and winter (*p* = 0.0136). The differences between winter and spring, as well as summer and autumn, showed no statistical significance (Figure 1).

The means ± standard deviations (SD) and medians of the mtDNA:nDNA ratio by seasons are shown in Figure 2. The mean ± SD of mtDNA:nDNA ratio was 271 (174) in summer, 274 (161) in autumn, 384 (174) in winter, 436 (193) in spring, and 349 (186) overall. Similarly, the median (minimum, maximum) of mtDNA:nDNA ratio was 246 (99.4, 888) in summer, 236 (71.5, 755) in autumn, 345 (90.8, 867) in winter, 366 (120, 920) in spring, and 315 (71.5, 920) in overall seasons. The mean mtDNA:nDNA ratio in winter was about 1.4-fold higher than in summer and autumn. While the mean mtDNA:nDNA ratio in spring was about 1.6-fold higher than in summer and autumn.

### 2.2. The Effect of Age on the Mitochondrial Mass (mtDNA:nDNA Ratio)

The means ± standard deviation (SD) and medians of mtDNA:nDNA ratios by age are shown in Figure 3. The mean ± SD of mtDNA:nDNA ratio was 356 (189) during young age, 390 (212) during midlife, 329 (164) during mature adult age, 334 (179) during late adult age, 296 (165) during old age, and 349 (186) during the overall life phase. Likewise, the median (minimum, maximum) of the mtDNA:nDNA ratio was 334 (91, 920) during young age, 325 (104, 867) during midlife, 329 (164) during mature adult age, 334 (179) during late adult age, 296 (165) during old age, and 349 (186) during the overall life phase. Our findings indicate that the amount of mtDNA:nDNA ratio is lower in elderly persons as compared to younger individuals. However, the *p*-value of the difference in the mtDNA:nDNA ratio between old and young people was 0.8 and between old and midlife was 0.3, indicating that aging was not a significant factor to influence the mitochondrial mass.

### 2.3. The Effects of Age, Season, and Mitochondrial Mass on Mitochondrial Performance (Respiration Per Mitochondrion)

To further explore the association between age, season, or mitochondrial mass and mitochondrial performance, we combine these findings with our Seahorse CSA data. To figure out whether the mitochondrial performance changes during aging, first, we compared the age groups under 50 and over 50 years with mitochondrial performance. The tendency for higher mitochondrial performance was observed during aging. However, the difference in mitochondrial performance in these age groups was not statistically significant (*p* = 0.075). The tremendous change in mitochondrial performance in the group over 50 years was caused by a marked increase from late adulthood to older persons (Figure 4).

The higher mitochondrial performance in older people was further confirmed by a significant correlation between mitochondrial performance and age (Figure 5). With every year of life, mitochondrial respiration increases by approximately 0.0114 pmol/min/mitochondrion, with a *p*-value of 0.017 and correlation coefficient (r) of 0.327.

In this study, we also examined the association between mitochondrial performance and seasonal variations. Mitochondrial performance was higher in summer (mean MP = 1.14) and autumn (mean MP = 1.09) compared to winter (mean MP = 0.735) and spring (mean MP = 0.680) (Table 1).

To figure out whether warm months display higher mitochondrial respiration than cold months, we created two groups. In the warm group, we collected the data from the warmest months in the year: June, July, and August. In the cold group, the coldest months were November, December, January, and February. We compared the mitochondrial performance in the two groups (Figure 6). The mitochondrial performance was significantly higher during the warm months than the cold months (*p* = 0.007). Interestingly, individuals in the upper-50-years-old age group showed a much higher mitochondrial performance in warm months but a tremendous decline in cold months (Figure 6B). The lower-50-years-old group had approximately constant mitochondrial respiration in warm and cold months (Figure 6B).

As shown in Figure 7, there was a clear negative correlation between the mitochondrial performance and the mtDNA:nDNA ratio. The estimated effect of the mtDNA:nDNA ratio on the mitochondrial performance was −0.002 pmol/min/mitochondrion, with a *p*-value of <0.0001 and a correlation coefficient of −0.612. From this observation, it can be concluded that higher mtDNA content does not necessarily lead to higher mitochondrial performance. The correlation diagram showed that when the mtDNA:nDNA ratio exceeds 350, the mitochondrial performance declines, and a high mtDNA:nDNA ratio is a predictor of poor mitochondria function. Our hypothesis was that an increase the mitochondrial number might increase in the maximal performance of the cell. However, our findings showed that there were no, or even negative, correlations between the mtDNA:nDNA ratio and maximal respiration (r = −0.12), spare capacity in % (r = −0.05), as well as spare capacity in pmol/min (r = −0.11).

As shown in Figure 8, the estimated effect of the basal respiratory quotient (RQ) on the mitochondrial performance was 0.412, with a *p*-value of 0.149 and a correlation coefficient of 0.2 (Figure 8A). In contrast, there was a significant negative correlation between the mitochondrial performance and the RQ after the addition of carbonyl cyanide 4-(trifluoromethoxy) phenylhydrazone (FCCP) (*p* = 0.0006). FCCP uncouples the mitochondrial proton gradient, which enables electrons to reduce oxygen. FCCP was used at the concentration of 3 µM (final concentration). The estimated effect of the RQ after the addition of FCCP on mitochondrial performance was −0.712, with a correlation coefficient of −0.454 (Figure 8B).

## 3. Discussion

In this study, we investigated the mitochondrial mass based on mtDNA:nDNA and mitochondrial performance in PBMCs and the influence of mitochondrial mass on the results of CSA. The mtDNA:nDNA ratio was significantly lower during summer and autumn compared to winter and spring. However, the mitochondrial performance was significantly higher in summer and autumn compared to winter and spring. Similarly, our previous study showed that maximal respiration and spare capacity decreased in winter and continuously increased from spring until early autumn [22]. The high number of mitochondria during wintertime might be an attempt by the cell to fulfill the bioenergetic requirements of the body. This attempt is probably due to an increased energy demand in winter but a lowered supply of vitamins, such as vitamin D [17,23]. In this case, the cells try to maintain their function during winter and springtime by increasing the number of mitochondria. It is convincing that the more mitochondria numbers the cell needs to maintain its function, the less efficient its ATP production.

This study also showed that the amount of mtDNA was lower in elderly persons as compared to younger individuals. However, the difference was not statistically significant. Similarly, it is often considered that the number of mtDNA declines by age [24]. Other reports also showed that genetic variations, either nuclear or mitochondrial, can also result in lower mitochondrial mass and are exacerbated by aging [6,7].

This study also showed a clear tendency to increase mitochondrial performance with age. However, this tendency was not statistically significant. This insignificant result might be due to the small number of study participants. The tremendous difference in mitochondrial performance in the age group over 50 years was caused by a marked increase from late adulthood to older individuals. This finding was further confirmed by a significant correlation between mitochondrial performance and aging. With every year of life, mitochondrial respiration increases significantly. In our previous study, a similar result was observed in the case of mitochondrial respiration [22]. However, other previous studies reported that mitochondrial respiratory function declines in various human tissues during the aging process [2,3]. This is thought to be caused, at least partly, by oxidative damage and mutation of mitochondrial DNA (mtDNA) in the somatic tissues of aged individuals [3].

Interestingly, in this study, a much higher mitochondrial performance was observed in the age group over 50 years old in warm months but a tremendous decline in cold months. This might be due to a higher number of mitochondria in cold months in older individuals. In our previous study [22], we reported that extracellular acidification and mitochondrial respiration increase with age due to the fact that a slower metabolism has more reducing equivalents and, therefore, higher maximal respiration when FCCP is added. The high extracellular acidification is a sign of ATP-generating mechanisms apart from the mitochondria. The higher extracellular acidification, with a concomitant high mitochondrial performance and the observed high maximal respiration or spare capacity, are likely the results of high amounts of reducing equivalents. In contrast, low extracellular acidification, high ATP amounts, and high mitochondrial respiration accompanied by a low mtDNA:nDNA ratio are signs of potentially healthy mitochondria.

The current study revealed a clear negative correlation between the mtDNA:nDNA ratio and the mitochondrial performance, as well as some of the Seahorse CSA parameters, such as maximal respiration, spare capacity in %, and spare capacity in pmol/min. This indicates that higher mtDNA content does not necessarily lead to higher mitochondrial performance. The observed marked decline in mitochondrial performance was when the mtDNA:nDNA ratio exceeds 350, which implies that a high mtDNA:nDNA ratio (>350) is a predictor of poor mitochondria function. A previous study showed that mitochondrial mass is related to the outbreak of symptoms in LHON [21]. The higher the number of mitochondrial DNA copies, the lower the detection of LHON-related symptoms.

We also observed that the respiratory quotient (RQ) was elevated after the addition of FCCP in winter, while a decline in mitochondrial performance was observed. In our previous publication, we reported a similar relation between extracellular acidification and potential mitochondrial performance [22]. This increase in RQ after the addition of FCCP in winter and a negative correlation with mitochondrial performance shows that PBMCs are challenged with bioenergetic problems during wintertime.

In this study, the difference in mitochondrial performance among male and female study participants was not statistically significant. This might be due to the small number of study participants. Up to now, it has been unclear whether sex as a biological variable influences the mitochondrial bioenergetic profile, especially in PBMCs. However, it is noteworthy that sex-specific differences have been reported in metabolic responses following acute injury, such as trauma, shock, and sepsis [25,26].

**Limitations:** This study was limited by the lack of analysis of the molecular mechanisms of changes in mitochondrial mass and cellular stress. However, the findings of this study are strengthened by using a molecular method, qRT-PCR, for the determination of mitochondrial mass and by assessing a wide range of cellular stress parameters in PBMCs.

## 4. Materials and Methods

### 4.1. Study Design, Period and Settings

A retrospective cross-sectional comparative study was conducted at the Magdeburg Molecular Detections (MMD) laboratory from October 2017 to February 2020. Patients and healthy individuals who visited hospitals in Germany, and blood samples sent to the MMD laboratory for mitochondrial mass determination and cellular stress assay during the study period were included in this study. The demographic data of study participants, such as age, gender, and season of blood sample collection, as well as PBMCS isolation methods and laboratory results, were collected from the laboratory records.

### 4.2. Study Participants

In total, 188 study participants (87 females and 101 males) were enrolled in this study. PBMCs from 108 individuals were isolated using Optiprep^®^ (Opti-isolated), and 80 were isolated via negative selection using the Robosep^TM^-S system by Stemcell^TM^ technologies (Robo-isolated). Of the 188 persons, 36 individuals were recruited in spring, 41 in summer, 35 in autumn, and 76 in winter. For age-dependent comparisons, we defined five groups: young (9 to 35 years), midlife (35 to 50 years), mature adulthood (50 to 60 years), late adulthood (60 to 70 years), and old (70 to 90 years). These five groups were further summarized into two groups: under 50 years (9 to 50 years) and over 50 years (50 to 90 years old).

### 4.3. Blood Sample Collection and PBMCs Isolation

Venous blood (up to 16 mL) was collected from each participant following an aseptic technique in a vacutainer tube with Citrate-phosphate-dextrose solution with adenine (CPDA) anticoagulant. The blood samples were transported to the MMD laboratory immediately, and PBMCs were isolated for the determination of mitochondrial mass and CSA within 24 h of blood collection. PBMCs were isolated either by density centrifugation or negative selection, using magnetic bead methods as previously described [22].

### 4.4. Cellular Stress Assay

The cellular stress assay (CSA) was conducted using the Seahorse XFp Analyzer (Agilent^TM^, Santa Clara, CA, USA) or the Seahorse XF96 Analyzer (Agilent^TM^, Santa Clara, CA, USA) according to the manufacturer’s protocol, with 250,000 PBMCs (100 µL) per well in triplicates, as previously described [22]. Briefly: The Seahorse-based CSA assay measurement is performed with sequential stepwise injections of various mitochondrial inhibitors. Basal respiration is measured at first and followed by the addition of Oligomycin to inhibit the mitochondrial ATP synthase. This leads to an interruption in electron transport and, thus, lower oxygen consumption. Then, the injection of carbonyl cyanide 4-(trifluoromethoxy) phenylhydrazone (FCCP) uncouples the mitochondrial proton gradient, enabling electrons to reduce oxygen again. The maximal respiration is measured after the addition of FCCP. Finally, the addition of Rotenone and Antimycin A stops the mitochondrial respiratory chain. This sequential procedure enables the measurement of CSA parameters, namely basal respiration, nonmitochondrial respiration, proton leak, reserve respiration capacity, also called spare capacity, maximal respiration, and extracellular acidification rate (Figure 9).

### 4.5. Measurement of Mitochondrial Mass

The mitochondrial mass was determined by measuring the mtDNA:nDNA ratio in PBMCs using quantitative real-time PCR (qRT-PCR). For qRT-PCR, PBMCs were centrifuged for 10 min at 300× *g* at room temperature. The supernatant was discarded, and the obtained cell pellet was diluted in PBS and stored for DNA isolation. The DNA isolation was carried out with 1 × 10^6^ cells using the DNeasy^®^ Blood and Tissue (Qiagen, Hilden, Germany).

qRT-PCR was performed in a Rotor-Gene Q MDx 2plex Platform (Qiagen, Hilden, Germany) using the LightCycler^®^ Multiplex DNA Master Mix (Roche, Basel, Switzerland). The following primers were used: The pair 5′-AATATTAAACACAAACACCTACCT-3′(mtDNA-8446-F) and 5′-TGGTTCTCAGGGTTTgTTATAA-3′(mtDNA-8525-R) used for amplification of the mitochondrial DNA, and the pair 5′-CGGAACCGCTCATTGCC-3′(ß-Actin-F) and 5′-ACCCACACTGTGCCCATCTA-3′(ß-Actin-R) used for amplifying the nuclear ß-actin gene. The detection was carried out with the probes 5′-6FAM-CCTCACCAAAGC+C+C+ATA--BBQ-3′ for mtDNA and 5′-6FAM-CCTCACCAAAGC+C+C+ATA--BBQ-3′ for ß-actin. Probes and primers were obtained from TIB Molbiol, Berlin, Germany. The obtained results were analyzed using the 2^∆∆ct^ method. Eventually, the ratio of mtDNA to ß-Actin expression is referred to as mtDNA:nDNA ratio or mitochondrial mass.

For comparison of mtDNA:nDNA and Seahorse CSA data, we used only Robo-isolated PBMCs, as we observed more reliable Seahorse CSA data in Robo-isolated PBMCs than Opti-isolated PBMC in our previous report [22]. A total of 53 individuals were enrolled for this comparison (28 females and 25 males). PBMCS from 43 individuals were analyzed in a Seahorse XFp Analyzer (Agilent^TM^, Santa Clara, CA, USA), and 10 were analyzed in a Seahorse XF96 Analyzer (Agilent^TM^, Santa Clara, CA, USA).

### 4.6. Determination of Mitochondrial Performance (MP)

The previous reports showed that 2.5 copies of mitochondrial DNA are present per mitochondrion [27]. Therefore, from the mtDNA:nDNA ratio, we calculated the number of mitochondria per cell. To know the number of cells responsible for the maximal respiration in our Seahorse measurements, we also calculated the maximal respiration per mitochondrion, also called mitochondrial performance (MP), using the following formula.
$$\mathrm{Mitochondrial}\;\mathrm{performance}\;(\mathrm{MP})=\frac{\mathrm{Maximal}\;\mathrm{respiration}\;[\frac{\mathrm{pmol}}{\min}]}{\mathrm{mtDNA}:\mathrm{nDNA}/2.5}$$

### 4.7. Statistical Analysis

Statistical analyses were performed with R (version 3.6.2) and RStudio (version 1.2.5033). The following packages were used: tidyverse, ggpubr, table1, formattable, summarytools, psych, MASS, gmodels, lsr, Hmisc, corrplot, and caret. Due to the non-normal distribution of the data, we used a One-way ANOVA with the Tukey post hoc test to examine whether the obtained effects were statistically significant. A *p*-value < 0.05 was considered significant.

## 5. Conclusions

This study revealed that the number of mitochondria per cell peaks during winter and declines until summer, while the mitochondria performance increases in summer and decreases in winter. The mtDNA:nDNA ratio showed a negative correlation with mitochondrial performance and some of the important CSA parameters. These findings imply that the diagnosis of mitochondrial function during winter could lead to false-positive results. In contrast, mitochondrial underperformance in summer could be masked by generally better mitochondrial performance in summer. Therefore, we recommend that researchers and clinicians consider the mitochondrial mass and season of the year for the correct interpretation of CSA results and the proper management of mitochondrial underperformance.

## Figures and Tables

**Figure 1 ijms-24-14824-f001:**
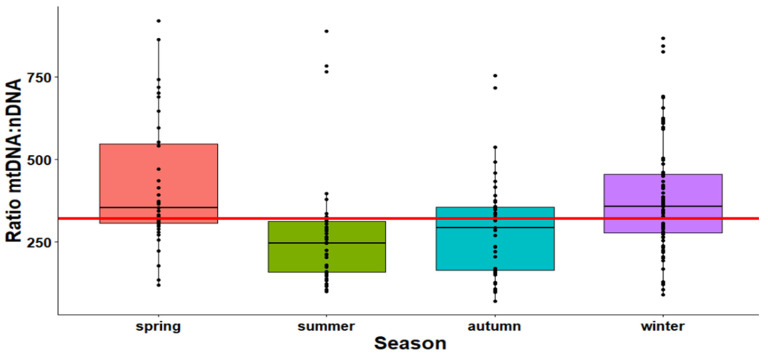
Comparison of the mtDNA:nDNA ratio by season in Opti- and Robo-isolated PBMCs. Spring (*n* = 36); summer (*n* = 41); autumn (*n* = 35); winter (*n* = 76). Spring = March, April, May; summer = June, July, August; autumn = September, October, November; winter = December, January, February. The red line indicates the median of all groups.

**Figure 2 ijms-24-14824-f002:**
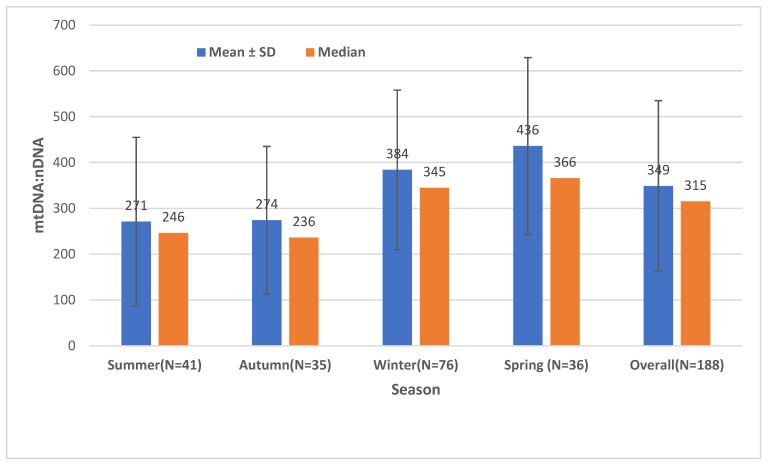
Mean ± standard deviation and median of the mtDNA:nDNA ratio by season in Robo- and Opti-isolated PBMCs (*n* = 188). Spring = March, April, May; summer = June, July, August; autumn = September, October, November; winter = December, January, February; SD = standard deviation.

**Figure 3 ijms-24-14824-f003:**
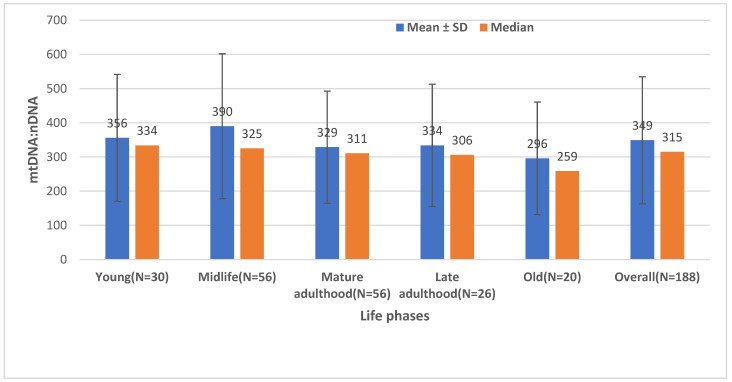
Mean ± standard deviation and median of mtDNA:nDNA ratio by life phase in Robo- and Opti-isolated PBMCs (*n* = 188). Young = 9 to 35 years; midlife = 35 to 50 years; mature adulthood = 50 to 60 years; late adulthood = 60 to 70 years; old = 70 to 90 years old; N = number, SD = standard deviation.

**Figure 4 ijms-24-14824-f004:**
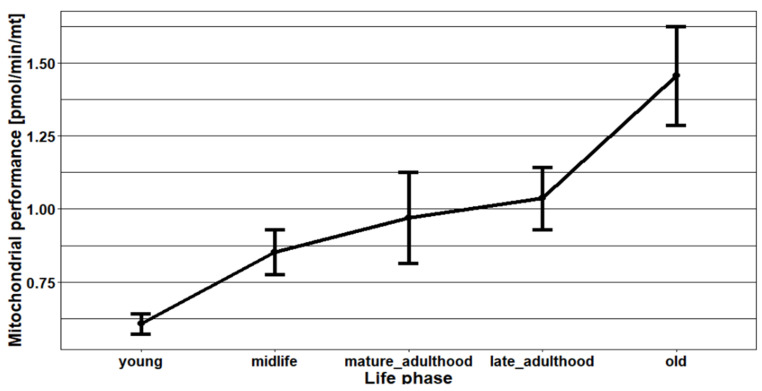
The changes in mitochondrial performance during aging. Young = 9 to 35 years (*n* = 3), midlife = 35 to 50 years (*n* = 18), mature adulthood = 50 to 60 years (*n* = 17), late adulthood = 60 to 70 years (*n* = 11), and old = 70 to 80 years old (*n* = 4). The black line shows the mean ± standard error of the mean (SE).

**Figure 5 ijms-24-14824-f005:**
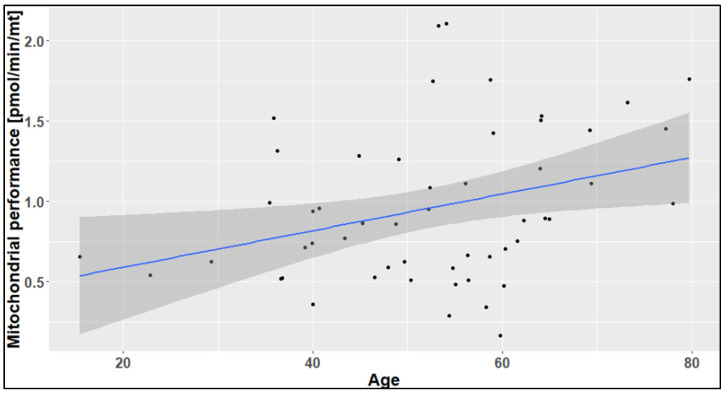
Correlation diagram of age against mitochondrial performance. The grey area around the regression line indicates the 95% confidence interval.

**Figure 6 ijms-24-14824-f006:**
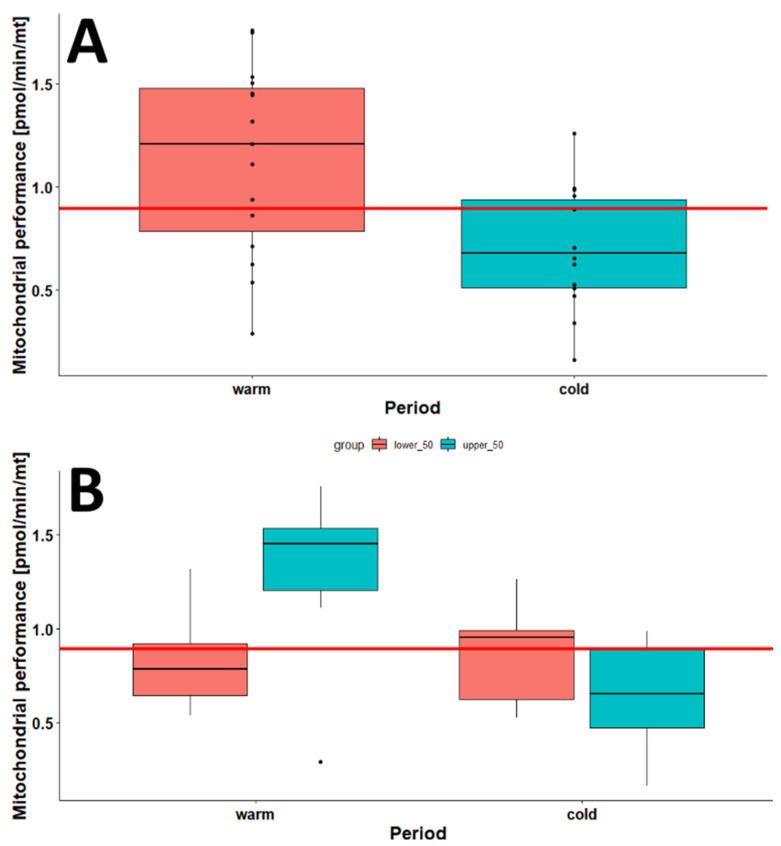
Comparison of the mitochondrial performance in Robo-isolated PBMCs analyzed in warmer months (warm: June, July, August; *n* = 15) and colder months (cold: November, December, January, February; *n* = 14). (**A**) Comparison of the mitochondrial performance in warm and cold months. (**B**) Comparison of warm and cold months depending on the age groups (lower-50 years, *n* = 5 in cold and *n* = 6 in warm; upper-50 years, *n* = 9 in cold and *n* = 9 in warm). The red line indicates the median of all groups.

**Figure 7 ijms-24-14824-f007:**
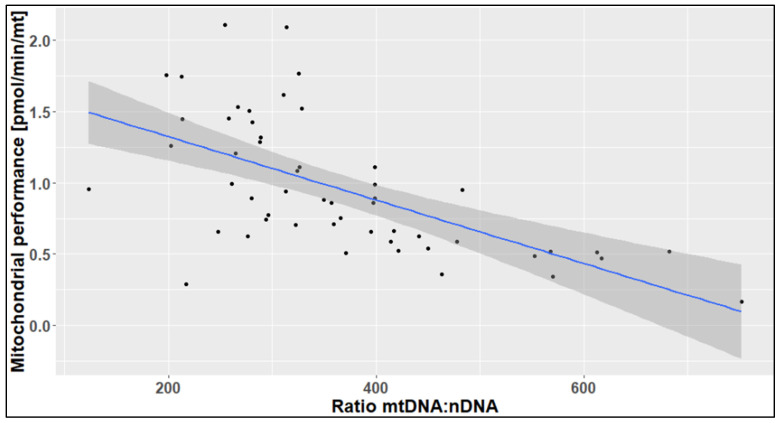
Correlation diagram of the mtDNA:nDNA ratio against mitochondrial performance. The grey area around the regression line indicates the 95% confidence interval.

**Figure 8 ijms-24-14824-f008:**
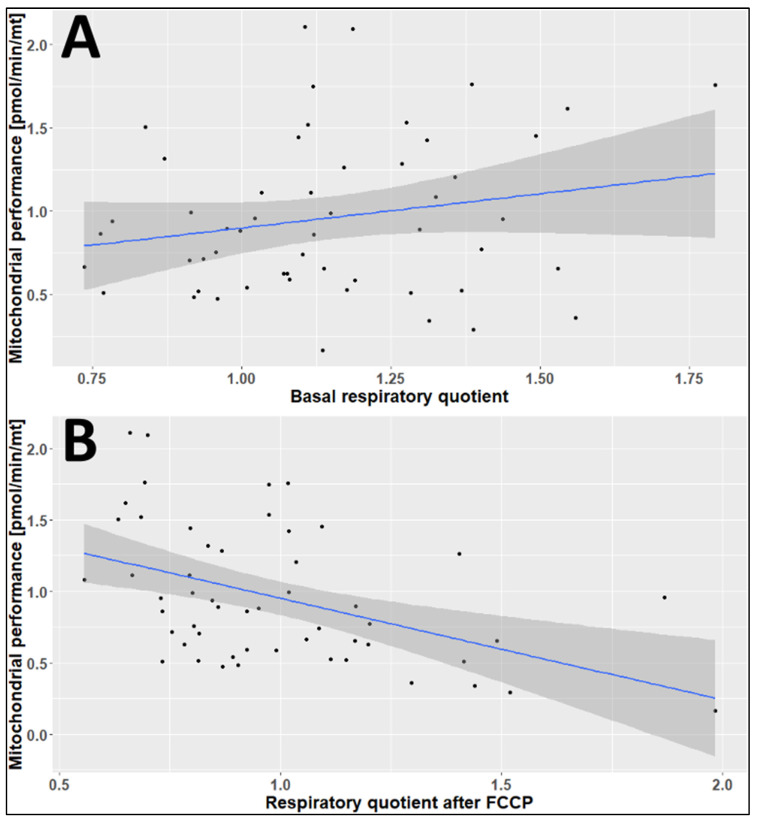
Correlation diagrams of the basal RQ (**A**) and the RQ after the addition of FCCP (**B**) against the mitochondrial performance. The grey area around the regression line indicates the 95% confidence interval.

**Figure 9 ijms-24-14824-f009:**
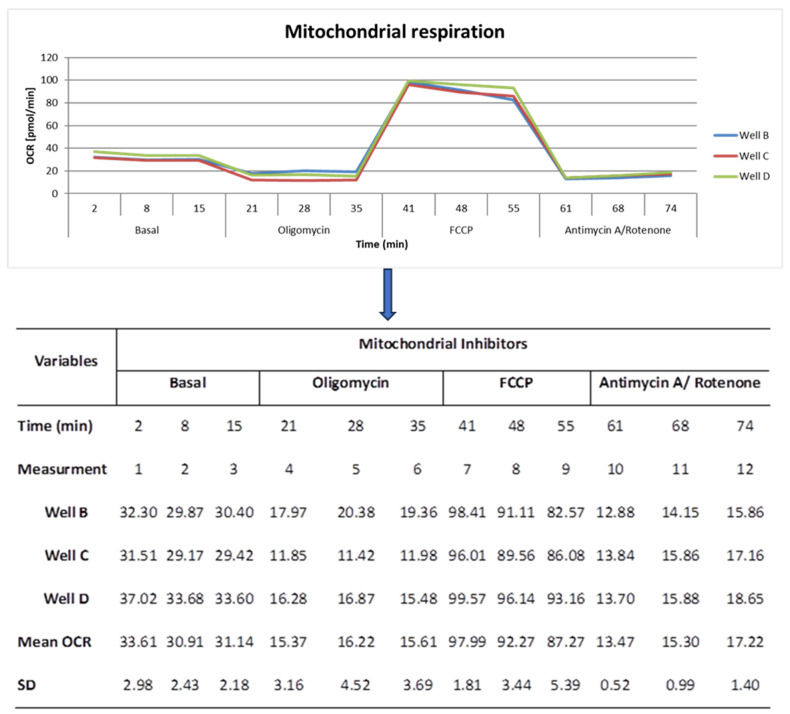
The evaluation procedure of the Seahorse oxygen consumption rate experiment in triplicates and the associated seahorse data curve, measurement results, mean, and standard deviation. SD = standard deviation; OCR = Oxygen consumption rate; FCCP = carbonyl cyanide 4-(trifluoromethoxy) phenylhydrazone; min = minute.

**Table 1 ijms-24-14824-t001:** Mean, SD, and Median of mitochondrial performance (MP) in Robo-isolated PBMC (*n* = 53).

Mitochondrial Performance	Spring	Summer	Autumn	Winter	Overall
	(*n* = 8)	(*n* = 15)	(*n* = 18)	(*n* = 12)	(*n* = 53)
Mean (SD)	0.680 (0.222)	1.14 (0.456)	1.09 (0.551)	0.735 (0.318)	0.960 (0.471)
Median [Min, Max]	0.588 (0.485, 1.11)	1.21 (0.290, 1.76)	0.905 (0.358, 2.11)	0.798 (0.164, 1.26)	0.882 (0.164, 2.11)

Spring = March, April and May; summer = June, July and August; autumn = September, October, November; winter = December, January, February; *n* = number; Min = minimum; Max = maximum; SD = standard deviation.

## Data Availability

All relevant data are included in this paper. The data are available on request from the corresponding author.

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
