# Peer review of "The Determination of Mitochondrial Mass Is a Prerequisite for Accurate Assessment of Peripheral Blood Mononuclear Cells’ Oxidative Metabolism"

_ijms, 2023, doi:10.3390/ijms241914824_

Round 1
Reviewer 1 Report
The article entitled “the determination of mitochondriaT mass is a prerequisite for adequate assessment of peripheral blood mononuclear cells oxidative metabolisme“ submitted by Belay Tessema, Riemer Janine, Ulrich Sack, Brigitte König is quite interesting but unfortunately badly presented. The authors should make some efforts for the presentation of the figures and improve elegant drawing. I find some similartity with the previous presentation of the same team (Tessema, B.; Riemer, J.; Sack, U.; König, B. Cellular Stress Assay in Peripheral Blood Mononuclear Cells: Factors Influencing Its Results. Int. J. Mol. Sci. 2022, 23, 13118), so the authors have to think about the way of writing and presenting the present data. There are some errors in numbering the tables. We noticed that not a single Seahorse measurement is presented. For the figure, I do not appreciate the presentation of the table. Please do somes nice histograms with ± SD and other histograms with the median values even if there is a wide variation. The statisticals analysis have to be explained.
At the present time the acceptation of this manuscript is questionnable since the amount of work is scarce and the presentation of low quality.
So, major revisions are required
Major comments
• Seahorse… Text.
When the authors say: “The application of Seahorse technology is a milestone in the diagnosis of mitochondrial dysfunctions”, I fully disagree since classical mitochondrial respiration is still conducted with O2 Clark electrode (a system that can eventually be modified to get the size of the mitochondrial membrane potential, with the introduction of a TPP+ or TMPD+ electrode and eventually set with pH electrode to measure the H+ extrusion linked to state-3 and state 4 situations (Gonzalvez et al. Cell Death and Differentiation (2005) 12, 614–626, in Figure 2 of the article). The High-Resolution Fluo Respirometry (O2k-Concept - Oroboros Instruments) is another instrument available. Each instruments had its particularity and flexibility. Part of the mitochondrial bioenergetic could also be measured by flow or image cytometry with different dues, i.e., for the measurement of the mitochondrial membrane potential, H2O2, O2.-superanions production.
Curiously the authors said: However, the findings of this study are strengthened by the use of qRT-PCR for the determination of mitochondrial mass. What does this mean?
The paragraph 5 conclusion is not at the correct place.
Minor comments
H2O2 should be presented as follow: H2O2
In Table 1 for better reading of the data. The mean / SD should be presented as Mean ± SD
The table 1 and 2 should be presented as histograms so the authors can put the ± SD or the median channel + variation on the histogram.
There is an error of presentation since we have two tables 1
Author Response
Reviewer Report 1
Review Report Form
Open Review
( ) I would not like to sign my review report
(x) I would like to sign my review report
Quality of English Language
( ) I am not qualified to assess the quality of English in this paper
( ) English very difficult to understand/incomprehensible
( ) Extensive editing of English language required
( ) Moderate editing of English language required
( ) Minor editing of English language required
(x) English language fine. No issues detected
|
Yes |
Can be improved |
Must be improved |
Not applicable |
|
|
Does the introduction provide sufficient background and include all relevant references? |
( ) |
(x) |
( ) |
( ) |
|
Are all the cited references relevant to the research? |
( ) |
(x) |
( ) |
( ) |
|
Is the research design appropriate? |
( ) |
(x) |
( ) |
( ) |
|
Are the methods adequately described? |
( ) |
(x) |
( ) |
( ) |
|
Are the results clearly presented? |
( ) |
(x) |
( ) |
( ) |
|
Are the conclusions supported by the results? |
( ) |
( ) |
(x) |
( ) |
Comments and Suggestions for Authors
Comment 1: The article entitled “the determination of mitochondrial mass is a prerequisite for adequate assessment of peripheral blood mononuclear cells oxidative metabolisme“submitted by Belay Tessema, Riemer Janine, Ulrich Sack, Brigitte König is quite interesting but unfortunately badly presented. The authors should make some efforts for the presentation of the figures and improve elegant drawing. I find some similartity with the previous presentation of the same team (Tessema, B.; Riemer, J.; Sack, U.; König, B. Cellular Stress Assay in Peripheral Blood Mononuclear Cells: Factors Influencing Its Results. Int. J. Mol. Sci. 2022, 23, 13118), so the authors have to think about the way of writing and presenting the present data. There are some errors in numbering the tables. We noticed that not a single Seahorse measurement is presented. For the figure, I do not appreciate the presentation of the table. Please do somes nice histograms with ± SD and other histograms with the median values even if there is a wide variation. The statisticals analysis have to be explained.
Response: Dear reviewer, we very much appreciate your review of our work and valuable comments and suggestions for improvement of our manuscript. We have revised our manuscript considering your comments and suggestions to improve the quality of our work.
We have made some efforts to improve the data presentation. Figure 2 and Figure 3.
We agree that there are some similarities with our previous publication regarding to the data presentation. Now we presented the data in this manuscript using additional two different figures.
The errors in numbering the tables were a mistake in writing the table titles, both tables had the same title. Now even though the tables are changed into figures as per your suggestion, the figures captions (titles) are correctly written. Figure 2 and Figure 3.
Yes, we used Seahorse XFp and Seahorse XF96 analyzers for CSA measurement. However, in our previous publication, CSA parameters measured by both Seahorse XFp and Seahorse XF96 analyzers demonstrated comparable results (ref. 22).
We have changed table 1 to histogram and the statistical analysis is explained. Figure 2.
Major comments
Comment 2: Seahorse… Text.
When the authors say: “The application of Seahorse technology is a milestone in the diagnosis of mitochondrial dysfunctions”, I fully disagree since classical mitochondrial respiration is still conducted with O2 Clark electrode (a system that can eventually be modified to get the size of the mitochondrial membrane potential, with the introduction of a TPP+ or TMPD+ electrode and eventually set with pH electrode to measure the H+ extrusion linked to state-3 and state 4 situations (Gonzalvez et al. Cell Death and Differentiation (2005) 12, 614–626, in Figure 2 of the article). The High-Resolution Fluo Respirometry (O2k-Concept - Oroboros Instruments) is another instrument available. Each instruments had its particularity and flexibility. Part of the mitochondrial bioenergetic could also be measured by flow or image cytometry with different dues, i.e., for the measurement of the mitochondrial membrane potential, H2O2, O2.-superanions production.
Response: We totally agree with your comment and your disagreement of our introducing sentence of the 3rd paragraph. We apologize for the unfavorable choice of words. We wanted to express that the Seahorse technology is a milestone for the entry of mitochondrial diagnostics into routine medical diagnostics.
Therefore, we changed the sentence “ The application of Seahorse technology is a milestone in the diagnosis of mitochondrial dysfunctions (11–13).” To “The Seahorse technology is a milestone in enabling mitochondrial bioenergetics to make its way into routine medical diagnostics with high throughput (11-13). Page 2, line 55-56.
Comment 3: Curiously the authors said: However, the findings of this study are strengthened by the use of qRT-PCR for the determination of mitochondrial mass. What does this mean?
Response: We have rephrased the sentence to make it clear as follow “However, the findings of this study are strengthened by using a molecular method, qRT-PCR for the determination of mitochondrial mass and by assessing a wide range of cellular stress parameters in PBMCs”. Page 12, Line 382-385.
Comment 4: The paragraph 5 conclusion is not at the correct place.
Response: To describe the important of this study, we have re-phrased this conclusion and keep it in this paragraph as “We believe that the analysis of the Seahorse based CSA is incomplete as long as the number of mitochondria actually contribute to the observed effects is unknown”. Page 2, line 81- 82.
Minor comments
Comment 5: H2O2 should be presented as follow: H2O2
Response: Corrected.
Comment 6: In Table 1 for better reading of the data. The mean / SD should be presented as Mean ± SD
Response: Corrected
Comment 7: The table 1 and 2 should be presented as histograms so the authors can put the ± SD or the median channel + variation on the histogram.
Response: Table1 and 2 are changed into Figure 2 and 3 as suggested.
Comment 8: There is an error of presentation since we have two tables 1
Response: The errors in numbering the tables were a mistake in writing the table titles, both tables had the same title. Now even though the tables are changed into figures as per your suggestion, the figures captions (titles) are correctly written. Figure 2 and Figure 3.
Reviewer 2 Report
The manuscript of Tessema et al. aims at determining the factors that are necessary for the adequate assessment of oxidative metabolism in isolated human peripheral mononuclear cells (PBMC). Authors enrolled 188 participants in the study and employed different methods to correlate PBMC mitochondrial performance with diverse other factors e.g. seasons, gae and mitochondrial mass. Authors applied cellular stress assay using the Seahorse equipment, determined the mitochondrial and nuclear DNA ration by quantitative RT-PCR and the mitochondrial performance by assessing maximal respiration rate relative to mitochondrial mass. Authors provide data demonstrating the number of mitochondria and mitochondrial performance present opposing seasonal variation. In addition, Authors suggest that mitochondrial performance is increased in older (70-80 years) individuals compared to persons of late adulthood age (60-70 years old). The data presented support the conclusions of the Authors. The conclusions draw attention to a scientific issue that is of interest of researchers in the oxidative stress field in general. Authors also include a paragraph addressing some of the perceived limitations of the study.
The manuscript is supported by 6 Figures and 3 Tables, and cites 25 references to put their finding in context of current knowledge. The manuscripts is well written, in a concise manner and easy to follow.
The manuscript fits the scope of the “International Journal of Molecular Sciences” and the manuscript is of interest for the readers of the journal.
This reviewer notes the following issues that need to be addressed before the manuscript could be considered for acceptation:
1. It would be interesting if other clinical data could be correlated with mitochondrial function e.g. body mass index as mitochondrial stress might reflect other metabolic pathologies (obesity in particular). If no such data can be obtained, Authors should mention literature that is available in this respect.
2. Is there any difference in the correlation between mitochondrial performance and age between male and female participants (Figure 3)? If statistical difference is not achievable due to low number of participants, Authors should at least mention some relevant literature describing sex differences in relation to mitochondrial function.
Minor issues:
3. Table 3 is marked Table 1
4. Different font types in Table 1 and in the Methods section, Page 10, section 4.4.
Author Response
Reviewer Report 2
Please provide a point-by-point response to the reviewer’s comments and either enter it in the box below or upload it as a Word/PDF file. Please write down "Please see the attachment." in the box if you only upload an attachment. An example can be found here.
Review Report Form
Open Review
(x) I would not like to sign my review report
( ) I would like to sign my review report
Quality of English Language
( ) I am not qualified to assess the quality of English in this paper
( ) English very difficult to understand/incomprehensible
( ) Extensive editing of English language required
( ) Moderate editing of English language required
( ) Minor editing of English language required
(x) English language fine. No issues detected
|
Yes |
Can be improved |
Must be improved |
Not applicable |
|
|
Does the introduction provide sufficient background and include all relevant references? |
(x) |
( ) |
( ) |
( ) |
|
Are all the cited references relevant to the research? |
(x) |
( ) |
( ) |
( ) |
|
Is the research design appropriate? |
(x) |
( ) |
( ) |
( ) |
|
Are the methods adequately described? |
(x) |
( ) |
( ) |
( ) |
|
Are the results clearly presented? |
(x) |
( ) |
( ) |
( ) |
|
Are the conclusions supported by the results? |
(x) |
( ) |
( ) |
( ) |
Comments and Suggestions for Authors
General Comment: The manuscript of Tessema et al. aims at determining the factors that are necessary for the adequate assessment of oxidative metabolism in isolated human peripheral mononuclear cells (PBMC). Authors enrolled 188 participants in the study and employed different methods to correlate PBMC mitochondrial performance with diverse other factors e.g. seasons, gae and mitochondrial mass. Authors applied cellular stress assay using the Seahorse equipment, determined the mitochondrial and nuclear DNA ration by quantitative RT-PCR and the mitochondrial performance by assessing maximal respiration rate relative to mitochondrial mass. Authors provide data demonstrating the number of mitochondria and mitochondrial performance present opposing seasonal variation. In addition, Authors suggest that mitochondrial performance is increased in older (70-80 years) individuals compared to persons of late adulthood age (60-70 years old). The data presented support the conclusions of the Authors. The conclusions draw attention to a scientific issue that is of interest of researchers in the oxidative stress field in general. Authors also include a paragraph addressing some of the perceived limitations of the study.
The manuscript is supported by 6 Figures and 3 Tables, and cites 25 references to put their finding in context of current knowledge. The manuscripts is well written, in a concise manner and easy to follow.
The manuscript fits the scope of the “International Journal of Molecular Sciences” and the manuscript is of interest for the readers of the journal.
Response: Dear reviewer, we are very much thankful for your meticulous review of our work and for providing us with very useful comments and suggestions. The manuscript has been revised according to your comments and suggestions to improve its quality.
This reviewer notes the following issues that need to be addressed before the manuscript could be considered for acceptation:
Comment 1. It would be interesting if other clinical data could be correlated with mitochondrial function e.g. body mass index as mitochondrial stress might reflect other metabolic pathologies (obesity in particular). If no such data can be obtained, Authors should mention literature that is available in this respect.
Response: We did not correlate other clinical data with mitochondrial function e.g. body mass index. It could be shown that the mitochondria of muscle and adipose tissue of subjects with obesity have altered bioenergetics when compared with mitochondria of lean subjects.
Heinonen S, Buzkova J, Muniandy M, Kaksonen R, Ollikainen M, Ismail K, et al. Impaired mitochondrial biogenesis in adipose tissue in acquired obesity. Diabetes 2015;64:3135–45. doi:10.2337/db14-1937.
Koliaki C, Roden M. Alterations of mitochondrial function and insulin sensitivity in human obesity and diabetes mellitus. Annu Rev Nutr 2016;36:337–67. doi:10.1146/annurev-nutr-071715-050656
However, with regard to peripheral blood mononuclear cells (PBMC) the group of Gomez didn`t find significant differences in maximal respiration and spare respiratory capacity between obese and lean subjects.
Angélica I. Borja-Magno, Janette Furuzawa-Carballeda, Martha Guevara-Cruz, Clorinda Arias, Julio Granados, Hector Bourges, Armando R. Tovar, Barry Sears, Lilia G. Noriega, Francisco Enrique Gómez,Supplementation with EPA and DHA omega-3 fatty acids improves peripheral immune cell mitochondrial dysfunction and inflammation in subjects with obesity, The Journal of Nutritional Biochemistry, Volume 120,2023,109415, ISSN 0955-2863, https://doi.org/10.1016/j.jnutbio.2023.109415. (https://www.sciencedirect.com/science/article/pii/S0955286323001481)
Comment 2: Is there any difference in the correlation between mitochondrial performance and age between male and female participants (Figure 3)? If statistical difference is not achievable due to low number of participants, Authors should at least mention some relevant literature describing sex differences in relation to mitochondrial function.
Response: We have mentioned the following paragraph and relevant literature describing sex differences in relation to mitochondrial function. “In this study, the difference in mitochondrial performance among male and female study participants was not statistically significant. This might be due to small number of study participants. Up to now it is unclear whether sex as a biological variable influences mitochondrial bioenergetic profile, especially in PBMCs. However, it is noteworthy that sex-specific differences have been reported in metabolic responses following acute injury, such as trauma, shock, and sepsis (26,27).” Page 12, Line 388- 393.
Minor issues:
Comment 3. Table 3 is marked Table 1
Response: corrected.
Comment 4. Different font types in Table 1 and in the Methods section, Page 10, section 4.4.
Response: Corrected.
Reviewer 3 Report
The authors have measured mtDNA/nDNA ratio, mito performance etc in individuals of different age groups and during different seasons. Although the results are rather interesting, the practical clinical implications and applications may not be generally feasible.
Here are some of my concerns:
1. The use of FCCP, the authors should give the full name of this chemicals, its mechanism of dissipating the mito potential and the concentration at which it was used.
2. Are the seasonal changes due to temperature alone, duration of daylight or both?
3. Related to the above point, how about the anticipated effects observed in countries near the equator, where seasonal changes are minimal, or in the southern hemispheres, where seasonal changes are opposite. The authors should discuss this and also find literature where similar studies were conducted.
4. I find the result that ageing increases mito performance to be intriguing. The authors should provide more molecular explanations and quote more studies on this topic and discuss whether this is a universal phenomenon in other ethnic groups.
5. The use of the term mito dysfunction in the Conclusion is rather misleading. I think unless the authors could provide evidence that there was depolarization of mito potential, mito Ca overload etc, the term dysfunction should not be used. Rather the term under-performance may be preferable.
need some revision: especially multiple occations there should be coma, instead of full stop.
Author Response
Reviewer Report 3
Review Report Form
Open Review
(x) I would not like to sign my review report
( ) I would like to sign my review report
Quality of English Language
( ) I am not qualified to assess the quality of English in this paper
( ) English very difficult to understand/incomprehensible
( ) Extensive editing of English language required
(x) Moderate editing of English language required
( ) Minor editing of English language required
( ) English language fine. No issues detected
|
Yes |
Can be improved |
Must be improved |
Not applicable |
|
|
Does the introduction provide sufficient background and include all relevant references? |
( ) |
(x) |
( ) |
( ) |
|
Are all the cited references relevant to the research? |
( ) |
(x) |
( ) |
( ) |
|
Is the research design appropriate? |
( ) |
(x) |
( ) |
( ) |
|
Are the methods adequately described? |
( ) |
(x) |
( ) |
( ) |
|
Are the results clearly presented? |
( ) |
(x) |
( ) |
( ) |
|
Are the conclusions supported by the results? |
( ) |
(x) |
( ) |
( ) |
Comments and Suggestions for Authors
General comment: The authors have measured mtDNA/nDNA ratio, mito performance etc in individuals of different age groups and during different seasons. Although the results are rather interesting, the practical clinical implications and applications may not be generally feasible.
Response: Dear reviewer, we are very much thankful for your review of our work and for providing us with very useful comments and suggestions. The manuscript has been revised according to your comments and suggestions to improve its quality.
The aim of this work was to show that individual mitochondrial performance is an important parameter for assessing mitochondrial bioenergetics. This requires the determination of mitochondrial mass. An increase in mitochondrial mass compensates for inadequate cellular energy production, at least at the beginning of mitochondrial dysfunction. An increase in mitochondrial mass influences cellular susceptibility to energy deficiency. An example is the investigations at LHON (21) and (Bianco A, Valletti A, Longo G, Bisceglia L, Montoya J, Emperador S, Guerriero S, Petruzzella V. Mitochondrial DNA copy number in affected and unaffected LHON mutation carriers. BMC Res Notes. 2018 Dec 20;11(1):911. doi: 10.1186/s13104-018-4025-y. PMID: 30572950; PMCID: PMC6302380.)
This study is intended to open the door to detailed studies of mitophagy and mitochondrial biogenesis in everyday clinical practice. In addition, we are convinced that determining individual mitochondrial performance in conjunction with cellular mitochondrial mass opens new therapeutic strategies to combat mitochondrial dysfunction.
Here are some of my concerns:
Comment 1. The use of FCCP, the authors should give the full name of this chemicals, its mechanism of dissipating the mito potential and the concentration at which it was used.
Response: We provided the detail information about FCCP as per you comment as followw: Carbonyl cyanide 4-(trifluoromethoxy) phenylhydrazone (FCCP) which uncouples the mitochondrial proton gradient, enabling electrons to reduce oxygen. FCCP was used at the concentration of 3µM (final concentration). Page 9, Line 288-291.
Comment 2. Are the seasonal changes due to temperature alone, duration of daylight or both?
Response: We agree with you that seasonal changes do not have to depend only on temperature. The length of daylight can also have an influence. It is also possible that the different irradiance levels from the sun (around 700 watts/square meter in summer; around 247 watts/square meter in winter in Central Europe) have an influence on the seasonal differences in mitochondrial function. We also pointed out the role of UV-B radiation intensity, which varies seasonally, for vitamin D3 formation.
Comment 3. Related to the above point, how about the anticipated effects observed in countries near the equator, where seasonal changes are minimal, or in the southern hemispheres, where seasonal changes are opposite. The authors should discuss this and also find literature where similar studies were conducted.
Response: We would like to have further insights into the dependence of mitochondrial bioenergetics on individual factors such as temperature, solar radiation intensity, electromagnetic pattern, air pollution, and many lifestyle factors such as diet and exercise. Our studies in Central Europe have clearly shown that there are seasonal factors influencing mitochondrial bioenergetics. We agree with the reviewer that further investigations, for example in countries near the equator or in the southern hemisphere, would provide significant knowledge. To our knowledge and research, there are no such studies to date. The possible influence of different mitochondrial haplotypes, which can be found particularly at the equator and in the southern hemisphere, should be considered.
Comment 4. I find the result that ageing increases mito performance to be intriguing. The authors should provide more molecular explanations and quote more studies on this topic and discuss whether this is a universal phenomenon in other ethnic groups.
Response: Aging is a complex process that involves a decline in biological and metabolic activity, resulting in the onset of various age-related conditions, such as neurodegenerative diseases, cardiovascular diseases, metabolic diseases, immune system diseases, and cancer. One key aspect of aging that has been extensively studied is its impact on mitochondrial function or vice versa. On the other hand, mitochondrial activity may not decline linearly throughout the aging process (Baker DJ, Peleg S. Biphasic Modeling of Mitochondrial Metabolism Dysregulation during Aging. Trends Biochem Sci. 2017 Sep;42(9):702-711. doi: 10.1016/j.tibs.2017.06.005. Epub 2017 Jun 29. PMID: 28669456.). Therefore, we agree that we need more studies on the molecular mechanism of mitochondrial functions in aging including different ethnic groups.
The aim of our study was not to elucidate the molecular mechanisms that lead to under-performance of mitochondrial function. With the fundamental results of our study, further investigations into the underlying molecular mechanisms can be followed.
Comment 5. The use of the term mito dysfunction in the Conclusion is rather misleading. I think unless the authors could provide evidence that there was depolarization of mito potential, mito Ca overload etc, the term dysfunction should not be used. Rather the term under-performance may be preferable.
Response: Corrected as suggested.
Comment 6: Comments on the Quality of English Language
need some revision: especially multiple occations there should be coma, instead of full stop.
Response: Corrected as suggested.
Round 2
Reviewer 1 Report
It is clear that the manuscript has been improved but stillthere is lot to do.
The authors have to provide some exemple of there seahorse measurements they have done
the standard deviation from the mean values are not notified on the graphs provided, this has to be done.
I am not sure that the work is very important indeed.
Author Response
Reviewer 1 (Round 2)
Review Report Form
Open Review
( ) I would not like to sign my review report
(x) I would like to sign my review report
Quality of English Language
( ) I am not qualified to assess the quality of English in this paper
( ) English very difficult to understand/incomprehensible
( ) Extensive editing of English language required
( ) Moderate editing of English language required
( ) Minor editing of English language required
(x) English language fine. No issues detected
|
Yes |
Can be improved |
Must be improved |
Not applicable |
|
|
Does the introduction provide sufficient background and include all relevant references? |
(x) |
( ) |
( ) |
( ) |
|
Are all the cited references relevant to the research? |
(x) |
( ) |
( ) |
( ) |
|
Is the research design appropriate? |
(x) |
( ) |
( ) |
( ) |
|
Are the methods adequately described? |
(x) |
( ) |
( ) |
( ) |
|
Are the results clearly presented? |
( ) |
( ) |
(x) |
( ) |
|
Are the conclusions supported by the results? |
( ) |
(x) |
( ) |
( ) |
Comments and Suggestions for Authors
General comment: It is clear that the manuscript has been improved but stillthere is lot to do.
Response: Dear reviewer, we are very much thankful for your review of our work for the second time and for providing us with very useful comments. The manuscript has been revised according to your comments to improve its quality.
Comment 1: The authors have to provide some exemple of there seahorse measurements they have done.
Response: Seahorse measurement is now one of the routine techniques, at least in research and now also in routine medical diagnostics. In recent years, thousands of publications have appeared that have integrated seahorse measurement. Among them around 1000 publications deal with immune cells including peripheral blood mononuclear cells. From our point of view, there is no added benefit for the reader if a standard seahorse measurement is shown. It would be different if, for example, the influence of medications or modulators of signaling pathways on mitochondrial function were the subject of our investigations.
Comment 2: the standard deviation from the mean values are not notified on the graphs provided, this has to be done.
Response: We have included standard deviations from the mean values on the graphs. Figure 2 and Figure 3.
I am not sure that the work is very important indeed.
Round 3
Reviewer 1 Report
I am not sure that the authors really answered my proper question.
They have modified the presnetation of the data but just forget to put the standard deviation...
I asked them to present some Oroboros data (curves) but they did not do any modification of the manuscript.
Since no proper answer is given... We are back to the same position...
So, at the present state the manuscript stayed of quite low quality and I am not sure that this acceptable for publication in the IJMS Journal
Author Response
Reviewer 1 Round 3
Open Review
( ) I would not like to sign my review report
(x) I would like to sign my review report
Quality of English Language
( ) I am not qualified to assess the quality of English in this paper
( ) English very difficult to understand/incomprehensible
( ) Extensive editing of English language required
( ) Moderate editing of English language required
( ) Minor editing of English language required
(x) English language fine. No issues detected
|
Yes |
Can be improved |
Must be improved |
Not applicable |
|
|
Does the introduction provide sufficient background and include all relevant references? |
(x) |
( ) |
( ) |
( ) |
|
Are all the cited references relevant to the research? |
(x) |
( ) |
( ) |
( ) |
|
Is the research design appropriate? |
( ) |
(x) |
( ) |
( ) |
|
Are the methods adequately described? |
(x) |
( ) |
( ) |
( ) |
|
Are the results clearly presented? |
(x) |
( ) |
( ) |
( ) |
|
Are the conclusions supported by the results? |
( ) |
(x) |
( ) |
( ) |
Comments and Suggestions for Authors
General Comment: I am not sure that the authors really answered my proper question.
Response: Dear reviewer, thank you so much for your review of our work again and for providing us with very useful suggestions. The manuscript has been revised according to your suggestions to improve its quality.
Comment 1. They have modified the presnetation of the data but just forget to put the standard deviation.
Response: We have included the standard deviation bar for each variable in both figures. Figure 2 and Figure 3
Comment 2: I asked them to present some Oroboros data (curves) but they did not do any modification of the manuscript.
Response: We have presented a Seahorse data curve (Figure 9) with texts describing the sequential stepwise measurement of Seahorse-based CSA parameters. Page 12 and 13, Line 423 – 455.
Since no proper answer is given... We are back to the same position...
So, at the present state the manuscript stayed of quite low quality and I am not sure that this acceptable for publication in the IJMS Journal
Round 4
Reviewer 1 Report
Still figure 2 and 3 do not have the ± SD draw on the draft.
And the authors did not provided us with an experimental Seahorse recording... with a comparison with two opposites samples for exmple, they did provide us witha theoritical scheme that explain the diffrent states of a Seahorse determination. But, I wanted an original record.
This manuscript will only be accepetd if this is provided
If not I decide REJECTION
Author Response
Reviewer 1 (Round 4)
Open Review
( ) I would not like to sign my review report
(x) I would like to sign my review report
Quality of English Language
( ) I am not qualified to assess the quality of English in this paper
( ) English very difficult to understand/incomprehensible
( ) Extensive editing of English language required
( ) Moderate editing of English language required
( ) Minor editing of English language required
(x) English language fine. No issues detected
|
Yes |
Can be improved |
Must be improved |
Not applicable |
|
|
Does the introduction provide sufficient background and include all relevant references? |
(x) |
( ) |
( ) |
( ) |
|
Are all the cited references relevant to the research? |
(x) |
( ) |
( ) |
( ) |
|
Is the research design appropriate? |
(x) |
( ) |
( ) |
( ) |
|
Are the methods adequately described? |
(x) |
( ) |
( ) |
( ) |
|
Are the results clearly presented? |
( ) |
( ) |
(x) |
( ) |
|
Are the conclusions supported by the results? |
(x) |
( ) |
( ) |
( ) |
Comments and Suggestions for Authors
Comment 1: Still figure 2 and 3 do not have the ± SD draw on the draft.
Response: Dear reviewer, we appreciate your repeated evaluation of our manuscript. We have included the ± SD draw on Figures 2 and 3. We hope that this change addresses the change you exactly wanted to be made in Figure 2 and 3. Figure 2 and Figure 3.
Comment 2: And the authors did not provided us with an experimental Seahorse recording... with a comparison with two opposites samples for exmple, they did provide us witha theoritical scheme that explain the diffrent states of a Seahorse determination. But, I wanted an original record.
Response: We have provided a figure with an experimental seahorse recording that shows the evaluation procedure of seahorse oxygen consumption rate experiment in triplicates and the associated seahorse data curve, measurement results, mean and standard deviation. Figure 9.